# Laser Powder Bed Fusion of 316L Stainless Steel: Effect of Laser Polishing on the Surface Morphology and Corrosion Behavior

**DOI:** 10.3390/mi14040850

**Published:** 2023-04-14

**Authors:** Jun Liu, Haojun Ma, Lingjian Meng, Huan Yang, Can Yang, Shuangchen Ruan, Deqin Ouyang, Shuwen Mei, Leimin Deng, Jie Chen, Yu Cao

**Affiliations:** 1Zhejiang Provincial Key Laboratory of Laser Processing Robotics, College of Mechanical & Electrical Engineering, Wenzhou University, Wenzhou 325035, China; 20461439087@stu.wzu.edu.cn; 2Sino-German College of Intelligent Manufacturing, Shenzhen Technology University, Shenzhen 518118, China; 2110412013@stumail.sztu.edu.cn (H.M.); 2110412020@stumail.sztu.edu.cn (L.M.); yangcan@sztu.edu.cn (C.Y.); ouyangdeqin@sztu.edu.cn (D.O.); 3Nantong Jinyuan Intelligent Technology Co., Nantong 226007, China; meishuwen@jyznjs.com; 4Wuhan National Research Center for Optoelectronics, Huazhong University of Science and Technology, Wuhan 430074, China; dlm@hust.edu.cn; 5Wenzhou University Rui’an Graduate College, Wenzhou University, Ruian 325207, China; chenjie@wzu.edu.cn

**Keywords:** selective laser melting, laser polishing, 316L stainless steel, surface morphology, electrochemical corrosion

## Abstract

Recently, laser polishing, as an effective post-treatment technology for metal parts fabricated by laser powder bed fusion (LPBF), has received much attention. In this paper, LPBF-ed 316L stainless steel samples were polished by three different types of lasers. The effect of laser pulse width on surface morphology and corrosion resistance was investigated. The experimental results show that, compared to the nanosecond (NS) and femtosecond (FS) lasers, the surface material’s sufficient remelting realized by the continuous wave (CW) laser results in a significant improvement in roughness. The surface hardness is increased and the corrosion resistance is the best. The microcracks on the NS laser-polished surface lead to a decrease in the microhardness and corrosion resistance. The FS laser does not significantly improve surface roughness. The ultrafast laser-induced micro-nanostructures increase the contact area of the electrochemical reaction, resulting in a decrease in corrosion resistance.

## 1. Introduction

Additive manufacturing (AM), as an advanced technology, has received increasing attention for its significant advantages in manufacturing high-strength and complex parts. Compared with machining and other traditional manufacturing technologies, this technology shows significant advantages, such as high flexibility, no molds, material saving, and reduced design cycle time [1,2,3]. It can manufacture digital models into solid parts directly by accumulating materials layer-by-layer. As a representative AM technique, laser powder bed fusion (LPBF) melts the metal powder in a predetermined scanning path by a high-power laser beam and then shapes it after cooling and solidification [4,5].

316L stainless steel is austenitic stainless steel. With its high strength and excellent corrosion resistance, 316L stainless steel has been widely used in aerospace, medical, food, and chemical industries. For the manufacture of 316L stainless steel, LPBF shows many advantages over traditional techniques, such as higher utilization of material, integrated forming, and the ability to create complex structures [6,7,8]. However, the parts manufactured by LPBF are prone to defects, such as unmelted powder, porosity, and cracks [9,10,11,12]. Moreover, due to the powder source material and layered manufacturing process, the surface roughness (SR) of the LPBF-ed 316L stainless steel is very rough, which cannot meet the requirements of aerospace and biomedical use [13,14,15]. Therefore, post-treatment is required to improve the surface quality of LPBF-ed parts. Currently, the technology for polishing metal parts includes grinding, sand-blasting, and electropolishing. However, these processes are still insufficient in polishing quality and efficiency for complex and thin-wall structures.

Recently, laser polishing, as an effective post-treatment technology, has received much attention for the advantages of non-contact, non-pollution, and high efficiency. With a continuous wave (CW) or long-pulse laser, this technology relies on laser-induced remelting and the subsequent rapid solidification of surface materials [16,17,18,19]. Previous studies show that for the additive manufacturing parts with initial roughness below 8 μm, pulsed lasers can reduce surface roughness by 71.3%, whereas CW lasers can reduce surface roughness by 84.4% [20,21]. Chen et al. used a CW laser to polish 316L stainless steel, reducing the surface roughness by 92%. Meanwhile, the microhardness and corrosion resistance could also be improved [22]. Pakin et al. studied the effect of processing parameters such as repetition frequency, laser power, and scanning speed on the polishing quality of aluminum alloys [23]. When the laser used for polishing is ultrafast, improved surface roughness is achieved by removing convex materials. Hafiz et al. adopted a picosecond laser to polish nickel-based alloy, and surface roughness could be reduced from 0.435 μm to 0.127 μm [24].

Although laser polishing with different pulse widths can reduce the roughness of metal surfaces, there are differences in the morphologies, metallographic structures, and mechanical properties of material surfaces. The CW laser can quickly obtain a flat surface, but the ultrafast laser can avoid the remelting and oxidation of surface materials. Currently, there is still no clear conclusion on the need to use lasers with different pulse widths for polishing metal components with different degrees of roughness.

In this paper, LPBF-ed 316L stainless steel with different initial surface roughness was polished using different pulse width lasers. The polishing efficiency, surface morphology, and phase composition achieved by the lasers with different pulse widths were analyzed. The cross-sectional microstructures and microhardness were investigated. In addition, the corrosion resistance characteristics of the laser-polished stainless steel surfaces were also evaluated. This work aims to achieve flexible and fast polishing of 3D-printed metal components with high roughness so that the prepared parts can meet the application requirements of most fields.

## 2. Experimental Procedures

### 2.1. Materials Preparation

The experimental raw material was commercially available 316L powder (Table 1). The powder morphology and size distribution are shown in Figure 1. The powder is spherical, which is favorable for LPBF forming (Figure 1a). The powder size range was 15–65 µm with 39.0 µm average diameter and 8.1 µm standard deviation (Figure 1b).

A batch of 15 × 15 × 15 mm^3^ cubic 316L stainless steel was made by LPBF equipment (SLM-100, Han’s Laser Co. Ltd., Shenzhen, China) under a nitrogen atmosphere. The LPBF process used a laser power of 180 W, a layer thickness of about 0.03 mm, and a scanning speed of 300 mm/s. The adjacent molding layers are scanned 67° apart in the direction of the LPBF process. Samples were cut from the substrate with an electric spark cutter and cleaned sequentially by sonication in ethanol and deionized water for 10 min to remove residual powder particles.

### 2.2. Laser Polishing Process

As shown in Figure 2b, the initial roughness of the LPBF-cube’s side surface (SS) is 16.28 μm, and the top surface (TS) roughness is 8.12 μm. To study the influence on low-roughness surfaces by laser polishing, the SS was pretreated with 80-grit sandpaper, and the SR was reduced to 0.97 μm. Such a low roughness made the surface look shiny. The pretreated SS was labeled P-SS. The adopted processing parameters are shown in Table 2, and the laser polishing with an area of 5 × 5 mm^2^ was carried out in an air environment (Figure 2). This study used the average surface roughness, Sa, to evaluate the laser polishing performance.

### 2.3. Microstructure and Mechanical Testing Method

The surface roughness of the samples was measured by a 3D laser confocal microscope (Olympus, OLS 5000), and the surface morphology of the samples was observed by scanning electron microscopy (SEM) (Carl Zeiss, GeminiSEM300, Oberkochen, Germany). The crystal phase of the samples was identified using X-ray diffraction (XRD) (SmartLab, Tokyo, Japan).

The microhardness of polished LPBF-ed 316L stainless steel samples was achieved using a nanoindentation method. The nanoindentation was carried out at a distance of approximately 20 μm from the sample cross-section, with 20 μm intervals each time, and the experimental results were the average of 10 measurements. The test was performed with a 500 mN load.

The electrochemical corrosion behaviors of samples were measured in 3.5% NaCl solution through an electrochemical workstation using a three-electrode cell. The measured 316L stainless steel sample, a saturated calomel electrode (SCE), and a platinum foil were used as working, reference, and counter electrodes. Open circuit potential (OCP) measurements were performed for 1800 s to ensure that the working surface reached a relatively stable state, and potentiodynamic polarization curves were obtained on OCP in the range of −1.5 V to +1.5 V with a scan rate of 0.5 mV/s. The impedance spectrum ranges from 10^−2^~10^5^ Hz with 10 mV AC signal amplitude. The following analytical tests were carried out on TS samples.

## 3. Results and Discussions

### 3.1. Effect of Surface Roughness and Pulse Width on Laser Polishing Performance

Figure 3a shows the polishing performances of the lasers with different pulse widths. For the P-SS with an initial Sa of 0.97 μm, the pulse width’s effect is not apparent. Moreover, after laser polishing, the changes in roughness are also small. The FS laser polishing only reduces the Sa by 0.12 μm (Table 3), and similar results are also achieved using the CW and NS lasers. However, the CW laser’s advantages come to the fore as the initial Sa increases. For the SS with the biggest Sa of 16.28 μm, the CW laser can quickly decrease the Sa to 1.15 μm, corresponding to a reduction of 92.9%. However, the Sa achieved by the NS laser is 3.11 μm, and the FS laser polished SS still possesses a Sa of 11.2 μm. Previous studies mainly focused on metal components with an initial roughness below 8 μm [15,20]. The maximum roughness of 16.28 μm is much higher than the previous research. With a high scanning speed of 300 mm/s, the CW laser can quickly reduce the roughness from 16.28 μm to about 1 μm. This reveals that the research result is very significant in polishing efficiency.

### 3.2. Effect of Laser Polishing on Microstructure

Figure 4 shows the surface morphologies of the samples before and after laser polishing. On the TS of the original LPBF-ed 316L stainless steel, some powder particles that are not entirely melted can still be observed (Figure 4a). Moreover, the laser-parallel-scanning-induced metal remelting is accompanied by sputtering. The above causes result in a ripple-like rough surface. The powder particles and ripples disappeared from the sample’s surface after CW laser polishing (Figure 4b). This is attributed to the remelting of materials (Figure 5). The laser-melted material flows by gravity and then re-solidifies, flatting the rough surface. However, the thermal stresses inside the material during the cooling process cause cracks on the polished surface (Figure 4b).

Compared to the CW laser-polished surface, the cracks become more obvious on the NS laser-polished surface (Figure 4b,c). This is because the NS laser has much less time to act on the material than the CW laser, resulting in a smaller heat conduction depth (Figure 4). The increased temperature difference along the normal direction of the material surface leads to an increase in thermal stress, making the crack more obvious. The FS laser polishing is based on removing convex materials (Figure 5). After three-laser scanning, the amount of material removal is very small (Figure 4d). Meanwhile, a new micro-nano structure is induced on the material surface, resulting in an insignificant reduction in roughness (Figure 3a). However, the cold processing effect caused by the ultrashort laser avoids the recast layer caused by the material melting, and the thermal-stress-induced microcracks are also inhibited (Figure 4d).

### 3.3. Surface XRD Analysis

Figure 6 shows the surface XRD patterns of the LPBF-ed 316L stainless steel samples before and after laser polishing. It can be seen that on the original sample, diffraction peaks with different intensities appear at 43.8°, 51.1°, and 74.9°, corresponding to face-centered cubic austenite (111), (200), and (220) crystal faces, respectively. After surface polishing with the three kinds of laser, no new diffraction peaks can be observed. This indicates that no new phase is generated on the polished surface, regardless of the pulse width of the laser used. This may be attributed to the high nickel content in 316L stainless steel, and the nickel can inhibit the formation of other phases [22].

### 3.4. Cross-Sectional Microstructure

Figure 7 exhibits the cross-sectional micrographs of original and laser-polished samples. The original LPBF-ed 316L stainless steel shows an uneven surface (Figure 7a). The semicircular metallographic organization caused by the unidirectional cooling of the melt pool is also clearly visible. After the CW laser polishing, the rough surface becomes flat, and a 49.08 μm remelted layer is formed (Figure 7b). Since the NS laser possesses a lower thermal effect, the remelted layer’s thickness is decreased to 36.53 μm (Figure 7c). The FS laser polishing mainly depends on removing convex material (Figure 7d), so the improvement in surface roughness is limited. However, the cold processing effect caused by the ultrashort laser prevents the material from remelting.

### 3.5. Cross-Sectional Microhardness Measurement

To evaluate the impact of different pulse width lasers on the specimens along the depth direction, a microhardness measurement was performed on the sample’s cross-sections. As shown in Figure 8, the average microhardness of the original LPBF-ed 316L stainless steel is about 3.11 Gpa, and the microhardness remained almost constant with increasing depth. The average hardness for the CW, NS, and FS laser-polished samples is 3.16 Gpa, 2.93 Gpa, and 3.04 Gpa, respectively. Figure 8a shows that the hardened layer of about 60 μm thickness is formed on the CW laser-polished sample. Related studies have shown that laser-polished additive manufactured parts produce gradient-hardened layers [25,26,27]. The cracks and thinned remelting layer for the NS laser-polished surface lead to decreased microhardness. Since there is no remelting layer on the FS laser-polished sample, the microhardness is almost constant along the depth direction.

### 3.6. Electrochemical Analysis

#### 3.6.1. Potentiodynamic Polarization Studies

The polarization curves of original and laser-polished samples are shown in Figure 9, and Table 4 shows the corresponding corrosion currents and potentials calculated by the Tafel extrapolation method. The higher the corrosion potential of the sample, the better the corrosion resistance. Moreover, the corrosion current determines the corrosion rate [28]. Surface roughness is essential to the material’s corrosion resistance [29]. The roughness reduction caused by CW laser polishing increases the corrosion potential and decreases the corrosion current. This indicates that the corrosion resistance is improved, and the corrosion rate is reduced [30]. The corrosion resistance of the NS laser-polished sample is not improved due to the surface cracks. During the electrochemical test, the electrolyte may penetrate the material along the cracks, resulting in corrosion inside the sample. The FS laser polishing does not obtain an improvement in surface roughness. Moreover, the ultrashort laser-induced micro-nanostructures increase the electrochemical reaction area, decreasing corrosion resistance. The polarization characteristics of the laser-polished P-SS and SS are consistent with the results shown in Figure 9 (Appendix A Figure A1). This indicates that the initial roughness does not affect the corrosion resistance of laser-polished samples with different pulse widths.

#### 3.6.2. Electrochemical Impedance Spectroscopic (EIS) Studies

The EIS spectra of the original and laser-polished samples are shown in Figure 10. Capacitive arcs can be observed on the Nyquist plots for all the samples (Figure 10a), which indicates that the corrosion reactions occurred at the stainless steel/electrolyte interface. The capacitive arc’s radius is essential for assessing corrosion resistance [31,32]. A large radius usually indicates excellent corrosion resistance. As shown in Figure 10a, the CW laser-polished sample’s radius is more significant than that of the other three samples, indicating the most vigorous corrosion resistance.

To further analyze the corrosion mechanism, an equivalent circuit model, shown in Figure 10d, was used to fit the impedance data. *R*_s_, *R*_f_, and *R*_ct_ represent the resistances of electrolyte solution, passivation film, and charge transfer, respectively. *Q*_f_ and *Q*_dl_ represent the passivation film capacitance and the double-layer capacitance. The fitted EIS parameter is shown in Table 5. The chi-squared values (*χ*^2^) ranged from 2.5 × 10^−5^ to 1.9 × 10^−3^, indicating a good agreement between the EIS data and the fitting results.

The corrosion rate *r* is inversely proportional to *R*_ct_ [33]:(1)$$R_{\mathrm{ct}}\times r=K$$

Since *K* is a constant, *R*_ct_ depends on the charge transfer rate caused by the Faraday process of redox reactions occurring on the electrode surface. As such, *R*_ct_ can be used to evaluate the corrosion resistance. As can be seen in Table 5, the CW laser-polished surface with the highest *R*_ct_ of 4.5 × 10^6^ Ω cm^2^ shows a lower corrosion rate than the other three samples. In addition, the CW laser-polished sample also has the lowest *Q*_f_ and the highest *R*_f_, which indicates a stable passivation film and high corrosion resistance [34,35].

#### 3.6.3. Electrochemical Corrosion Morphology

Figure 11 shows the corrosion micromorphologies of four samples after the EIS tests. Some corrosion pits can be observed on the original surface of the LPBF-ed 316L stainless steel (Figure 11a). Compared to the original sample, the corrosion pits on the CW laser-polished specimen are significantly reduced. Only a few micro-sized corrosion pores can be observed (Figure 11b). The corrosion pits on the NS laser-polished sample are mainly due to the surface cracks (Figure 11c). A large number of bumps on the FS laser-polished sample leads to the incompleteness of the passivation film during the corrosion process. The NaCl solution tends to corrode the material interior along these surface defects, resulting in a severely corroded surface (Figure 11d).

#### 3.6.4. Corrosion Mechanism

The corrosion mechanism of the original and laser-polished 316L stainless steel samples is shown in Figure 12. The oxidation reaction occurs at the anode:Fe − 2e^−^ → Fe^2+^(2)
Cr − 2e^−^ → Cr^2+^(3)

The reduction reaction occurs at the cathode:O_2_ + 2H_2_O + 4e^−^ → 4OH^−^(4)

The metal cations produced by the anodic reaction readily react with Cl^−^ in the NaCl solution to form metal chlorides, resulting in the continued dissolution of the metal in the solution:Cr^2+^ + 2Cl^−^ → CrCl_2_(5)
Fe^2+^ + 2Cl^−^ → FeCl_2_(6)

Next, the unstable CrCl_2_ and FeCl_2_ are easily transformed into Fe_2_O_3_ and Cr_2_O_3_ oxides due to the oxygen dissolved in the solution. The passivation film formed by the metal oxides can slow down the corrosion process of the material surface. However, the presence of surface microcracks leads to the erosion of Cl^−^ inside the material, promoting the formation of corrosion pits [36].

In summary, the rough structure on the original and FS laser-polished surface destroyed the integrity of the passivation film, leading to the formation of corrosion pits. The microcracks on the NS laser-polished sample also damaged the passivation film, resulting in a severely corroded surface. In contrast, the CW laser polishing could remove most of the defects on the LPBF-ed 316L stainless steel and obtain a flat surface, improving the corrosion resistance [37,38].

## 4. Conclusions

In this study, the surface morphologies and corrosion behaviors of LPBF-ed 316L stainless steel polished with different laser pulse widths were investigated.

(1) Although the NS laser can also significantly reduce the surface roughness, the generated microcracks decrease the microhardness and corrosion resistance.

(2) The FS can avoid the formation of remelting layers, but its improvement in the roughness is very limited. Moreover, the ultrashort pulse laser-induced micro-nanostructures increase the contact area of the electrochemical reaction, resulting in a decrease in corrosion resistance.

(3) The demonstrated advantage of the CW laser over the FS and NS laser is decided by the initial roughness. The surface material’s sufficient remelting realized by the CW laser can significantly improve the roughness. When the initial roughness of LPBF-ed stainless steel is higher than 10 μm, the CW laser can quickly reduce the surface roughness of LPBF-ed stainless steel to about 1 μm. Meanwhile, the surface hardness and corrosion can also be improved. The research results suggest that CW laser polishing presents promising applications in the pretreatment of LPBF-ed parts for ocean engineering, mechanical transmission, and punching die. However, compared to electrochemical polishing, the roughness of the laser-polished surface is still high. For applications requiring a surface roughness below 500 nm, the laser-polished surface must be treated using chemical and mechanical methods to further reduce the roughness.

## Figures and Tables

**Figure 1 micromachines-14-00850-f001:**
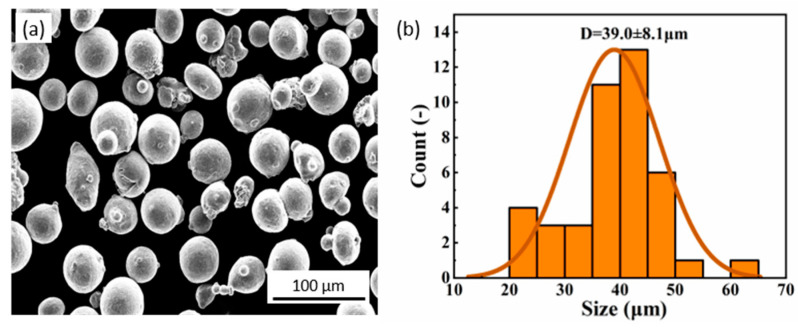
(**a**) SEM image and (**b**) size distribution of 316L stainless steel powders.

**Figure 2 micromachines-14-00850-f002:**
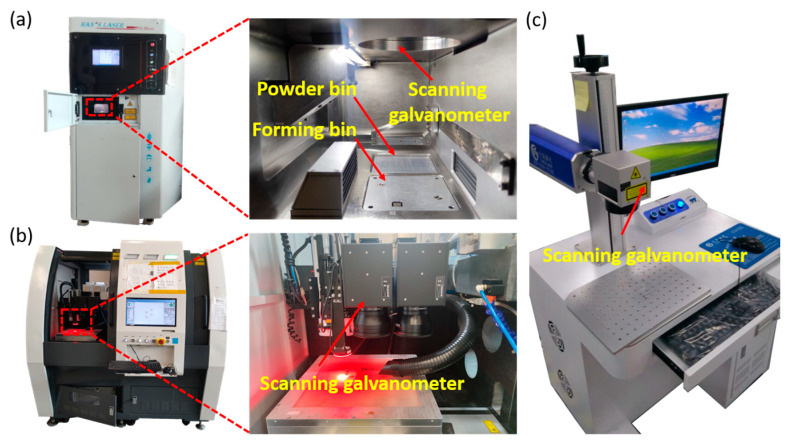
Actual images of the (**a**) CW, (**b**) FS, and (**c**) NS laser processing systems.

**Figure 3 micromachines-14-00850-f003:**
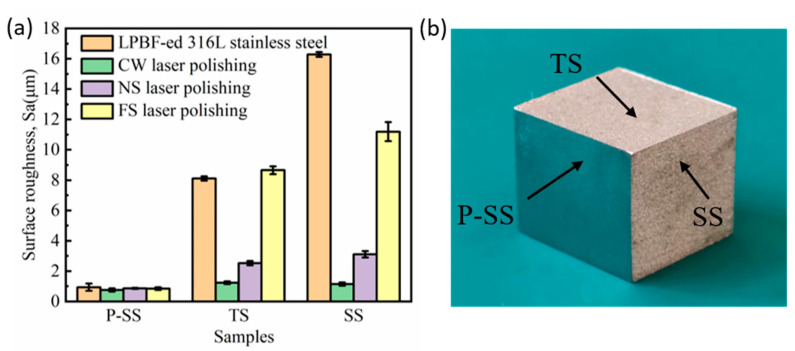
(**a**) Surface roughness of the polished samples with different laser pulse widths. (**b**) Object picture of the LPBF-ed 316L stainless steel.

**Figure 4 micromachines-14-00850-f004:**
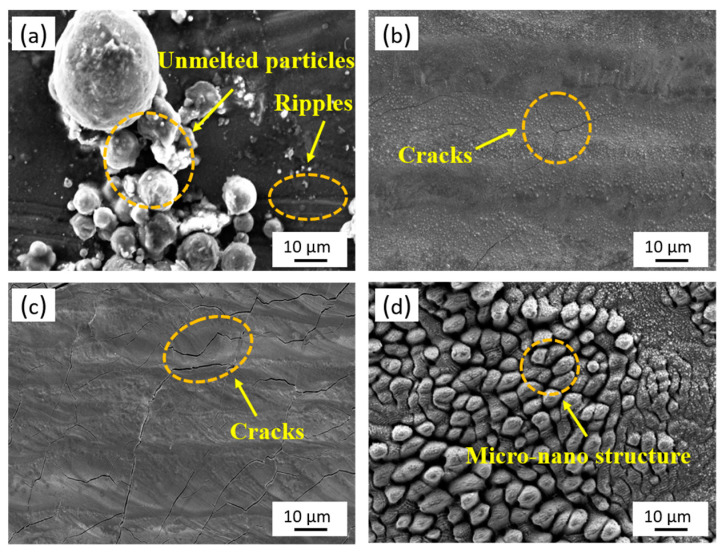
Surface morphologies of the (**a**) original, (**b**) CW, (**c**) NS, and (**d**) FS laser-polished 316L stainless steel surfaces.

**Figure 5 micromachines-14-00850-f005:**
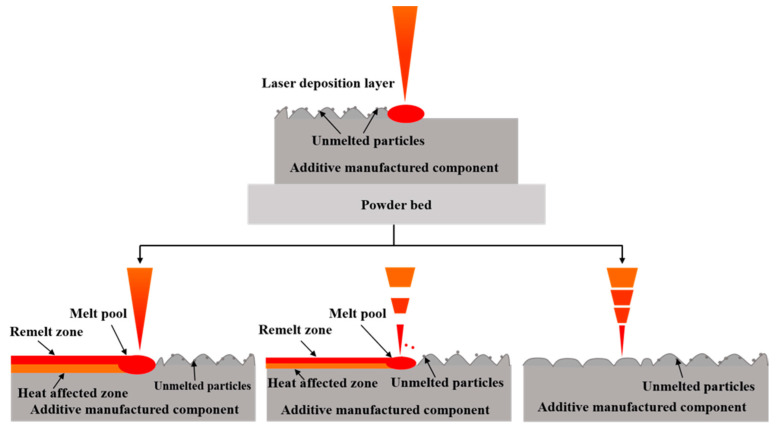
Schematic illustrating the laser polishing with different pulse widths.

**Figure 6 micromachines-14-00850-f006:**
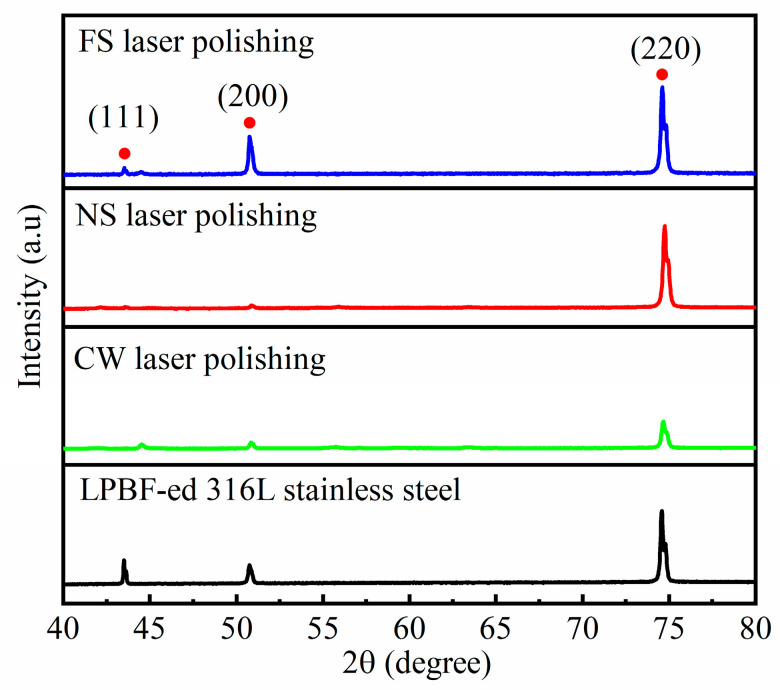
XRD patterns of the LPBF-ed 316L stainless steel before and after laser polishing.

**Figure 7 micromachines-14-00850-f007:**
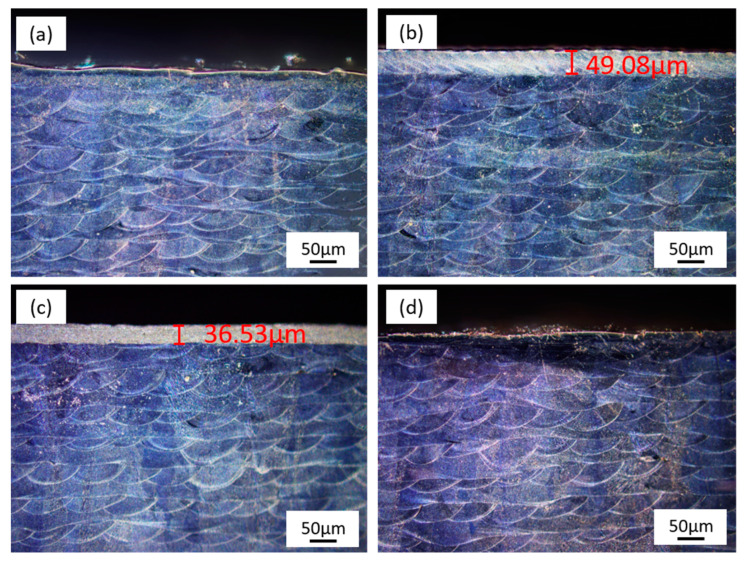
Cross-sectional micrographs of the (**a**) original, (**b**) CW, (**c**) NS, and (**d**) FS laser-polished 316L stainless steel surfaces.

**Figure 8 micromachines-14-00850-f008:**
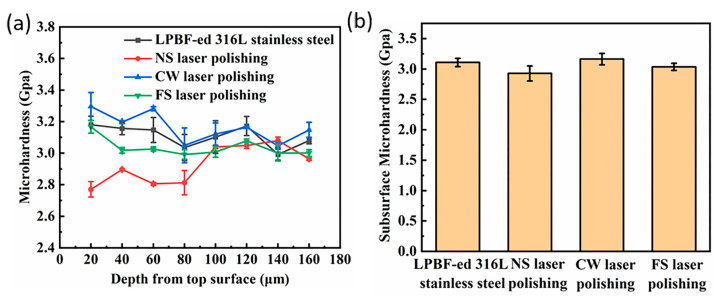
(**a**) Microhardness distributions and (**b**) average hardness of original and laser-polished samples.

**Figure 9 micromachines-14-00850-f009:**
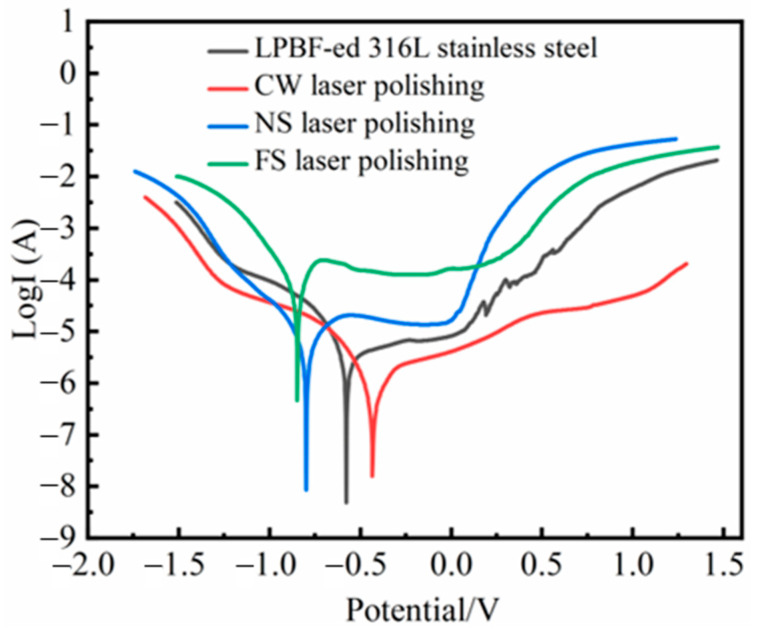
Potentiodynamic polarization curves of original and laser-polished samples.

**Figure 10 micromachines-14-00850-f010:**
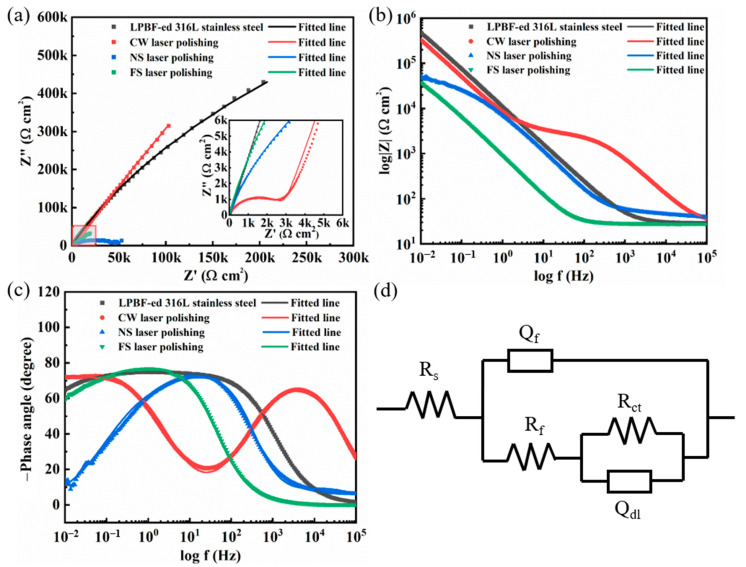
Measured and simulated (**a**) Nyquist curves, (**b**) Bode impedance, and (**c**) Bode phase angle of original and laser-polished samples. (**d**) Electrochemical equivalent circuit model.

**Figure 11 micromachines-14-00850-f011:**
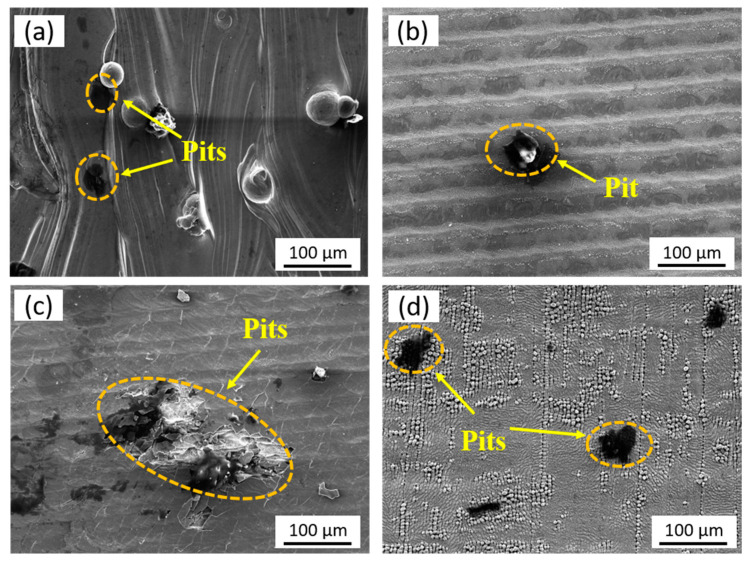
Surface corrosion morphologies of the (**a**) original, (**b**) CW, (**c**) NS, and (**d**) FS laser-polished 316L stainless steel samples.

**Figure 12 micromachines-14-00850-f012:**
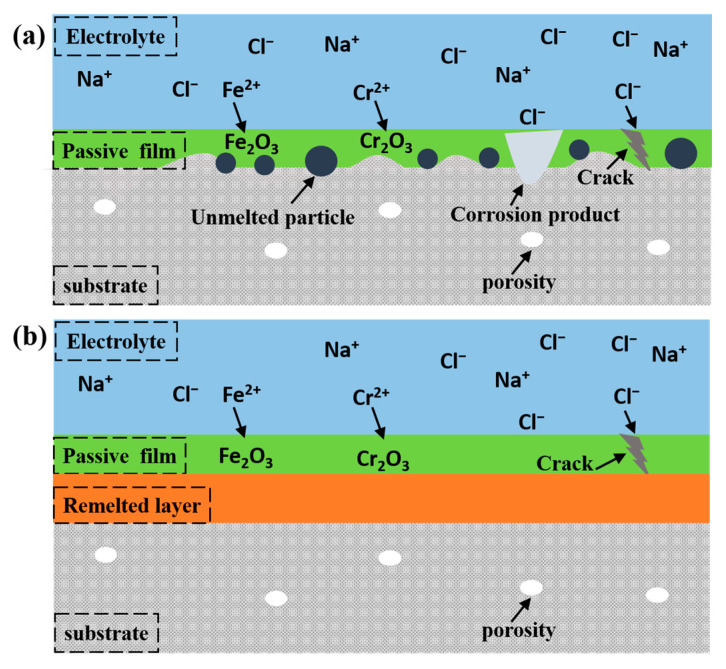
Corrosion mechanism diagram of the (**a**) original and (**b**) laser-polished 316L stainless steel.

**Table 1 micromachines-14-00850-t001:** Chemical composition of the 316L stainless steel powder.

Element	Ni	Cr	Mo	C	Mn	Si	Fe
Percent	10.72	16.96	2.44	0.01	0.73	0.51	balance

**Table 2 micromachines-14-00850-t002:** Parameters adopted for laser polishing.

	CW Laser	NS Laser	FS Laser
Energy density (J/mm^2^)	4.17	0.69	0.076
Scanning speed (mm/s)	300	100	500
Pulse frequency (kHz)	-	40	250
Number of Passes (-)	3	2	3

**Table 3 micromachines-14-00850-t003:** Surface roughness of the polished samples with different laser pulse widths.

	Sa (μm)		
Samples	P-SS	TS	SS
LPBF-ed 316L stainless steel	0.97	8.12	16.28
CW laser polishing	0.76	1.24	1.15
NS laser polishing	0.86	2.53	3.11
FS laser polishing	0.85	8.66	11.20

**Table 4 micromachines-14-00850-t004:** Quantitative information of four samples from the potentiodynamic polarization curves.

Samples	Corrosion Potential (V)	Corrosion Current (A)
LPBF-ed 316L stainless steel	−0.577	−7.181
CW laser polishing	−0.432	−7.722
NS laser polishing	−0.797	−6.909
FS laser polishing	−0.847	−6.038

**Table 5 micromachines-14-00850-t005:** EIS fitting results of an equivalent circuit model for original and laser-polished samples.

Sample	*R*_S_ (Ω cm^2^)	*Q* _f_	*R*_f_ (Ω cm^2^)	*Q* _dL_	*R*_ct_ (Ω cm^2^)	*χ* ^2^
LPBF-ed 316L stainless steel	29.14	8.67 × 10^−6^	74.05	7.40 × 10^−6^	4.13 × 10^6^	2.5 × 10^−5^
CW laser polishing	27	9.86 × 10^−7^	2900	3.14 × 10^−6^	4.5 × 10^6^	9 × 10^−4^
NS laser polishing	34.37	7.27 × 10^−5^	25.15	3.20 × 10^−5^	49,955	1.9 × 10^−3^
FS laser polishing	24.53	3.91 × 10^−5^	3.776	1.98 × 10^−4^	554,950	1.0 × 10^−4^

## Data Availability

Not applicable.

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
