# Peer review of "Laser Powder Bed Fusion of 316L Stainless Steel: Effect of Laser Polishing on the Surface Morphology and Corrosion Behavior"

_micromachines, 2023, doi:10.3390/mi14040850_

Round 1
Reviewer 1 Report
The author has well reported the Laser Powder Bed Fusion of 316L stainless steel and investigated the effect of laser polishing on the surface morphology and corrosion behavior. However, minor revision is suggested to the author.
1. It would be more interesting if the author could have provided the actual image of the experimental setup. Therefore, the author is suggested to provide the actual image of the experimental setup to make it more appropriate for the reader.
2. The reviewer is interested to know on what basis the author has identified the values of energy density, scanning speed, and pulse frequency while attempting the CW laser, NS laser, and FS laser. Justify?
3. In Figure 3, the text written in the SEM image is difficult to read and seems to be inappropriate from the reader's point of view. The author is suggested to use an arrow mark in the SEM image, whereas text can be added in the Figure-caption.
4. Figure 7 seems to be of poor quality and looks inappropriate from the reader's point of view. The author is suggested to improve the quality of the figure.
5. The author is strongly advised to mention the future scope of the present study.
6. The author is strongly advised to mention the limitation of the current research work in the conclusion section.
Reviewer 2 Report
I read this paper and the topic is interesting to publish. Please follow the comments.
1. What is the main issue that will be solved by this investigation? Please clarify it in the text.
2. Figure 1 has a shiny surface for P-SS. Please comment on this.
3. Why did you select Sa rather than Ra?
4. What does it add to the subject area compared with other published literature?
5. Please briefly introduce the process in the introduction.
6. What is the resolution of this work?
7. Double-check the wording of the paper. There are some mistakes and grammatical notes that need to be fixed.
8. There are several advantages of additive manufacturing over machining and other conventional manufacturing technologies. This needs to be bolded in your paper. Read and comprehend the following paper and add it to the introduction. “Laser subtractive and laser powder bed fusion of metals: a review of process and production features”.
Reviewer 3 Report
The current manuscript investigates the effect of laser polishing on the surface morphology of additively manufactured 316 stainless steel samples. the presented results are interesting and the experimental procedures are well written. However, some issues should be considered as follows:
- The introduction should include a more robust critical review to focus on the literature research related to surface roughness improvement of additively manufactured parts. Recent and significant references should be cited.
- The problem statement and the outline of the current work should be well defined by the end of introduction.
- What is the effect of using the sand paper before laser polishing? and how this step be applied for the complex shape parts? please add the illustration of these issues in the revised manuscript. can on the dimensional accuracy
- The obtained results and analysis should be validated and justified using the relationship between the performed measurements and citing of suitable references. For example, the relation between the XRD and microhardness measurements could be used to validate the behaviour of the microhardness values change.
- The conclusion should be focused including the main results, contributions, and novelty of the current work. A bullet points style is recommended to be used.
Round 2
Reviewer 2 Report
The authors made the paper ready to publish.
Reviewer 3 Report
The revised manuscript is significantly improved. The review comments and recommendations are well addressed.